

# Public transit mobility as a leading indicator of COVID-19 transmission in 40 cities during the first wave of the pandemic

Jean-Paul R. Soucy[1], Shelby L. Sturrock[1], Isha Berry[1], Duncan J. Westwood[2], Nick Daneman[2,3,4], David Fisman[1], Derek R. MacFadden[5] and Kevin A. Brown[1,4,6]

[1] Division of Epidemiology, Dalla Lana School of Public Health, University of Toronto, Toronto, Ontario, Canada
[2] Sunnybrook Health Sciences Centre, Toronto, Ontario, Canada
[3] Department of Medicine, University of Toronto, Toronto, Ontario, Canada
[4] ICES, Toronto, Ontario, Canada
[5] Ottawa Hospital Research Institute, Ottawa, Ontario, Canada
[6] Public Health Ontario, Toronto, Ontario, Canada

Corresponding author
Jean-Paul R. Soucy,
jeanpaul.soucy@mail.utoronto.ca

## ABSTRACT

**Background:** The rapid global emergence of the COVID-19 pandemic in early 2020 created urgent demand for leading indicators to track the spread of the virus and assess the consequences of public health measures designed to limit transmission. Public transit mobility, which has been shown to be responsive to previous societal disruptions such as disease outbreaks and terrorist attacks, emerged as an early candidate.

**Methods:** We conducted a longitudinal ecological study of the association between public transit mobility reductions and COVID-19 transmission using publicly available data from a public transit app in 40 global cities from March 16 to April 12, 2020. Multilevel linear regression models were used to estimate the association between COVID-19 transmission and the value of the mobility index 2 weeks prior using two different outcome measures: weekly case ratio and effective reproduction number.

**Results:** Over the course of March 2020, median public transit mobility, measured by the volume of trips planned in the app, dropped from 100% (first quartile ($Q_1$)–third quartile ($Q_3$) = 94–108%) of typical usage to 10% ($Q_1$–$Q_3$ = 6–15%). Mobility was strongly associated with COVID-19 transmission 2 weeks later: a 10% decline in mobility was associated with a 12.3% decrease in the weekly case ratio (exp ($\beta$) = 0.877; 95% confidence interval (CI): [0.859–0.896]) and a decrease in the effective reproduction number ($\beta$ = −0.058; 95% CI: [−0.068 to −0.048]). The mobility-only models explained nearly 60% of variance in the data for both outcomes. The adjustment for epidemic timing attenuated the associations between mobility and subsequent COVID-19 transmission but only slightly increased the variance explained by the models.

**Discussion:** Our analysis demonstrated the value of public transit mobility as a leading indicator of COVID-19 transmission during the first wave of the pandemic in 40 global cities, at a time when few such indicators were available. Factors such as persistently depressed demand for public transit since the onset of the pandemic limit the ongoing utility of a mobility index based on public transit usage. This study

illustrates an innovative use of "big data" from industry to inform the response to a global pandemic, providing support for future collaborations aimed at important public health challenges.

## INTRODUCTION

The COVID-19 pandemic, caused by the virus SARS-CoV-2, emerged in Wuhan, China in late 2019 and had spread around the globe by early March 2020. In response, governments implemented policies to limit transmission of the virus by reducing contacts between individuals outside their households, such as imposing limits on gatherings and closing shops, restaurants, and other venues (*Hale et al., 2021*). During these initial stages of the pandemic, there was a critical need for metrics capable of serving as leading indicators of community spread and for evaluating the effectiveness of interventions implemented to contain it. Early research pointed to human mobility data as a potential solution (*Buckee et al., 2020*; *Kraemer et al., 2020*). While commercial mobility datasets (*e.g.*, SafeGraph, Veraset, and X-Mode) had been available for purchase for some time, metrics relevant to the public at large were not yet widely accessible. Addressing this gap, Citymapper, a public transit app, launched a mobility index in mid-March 2020, based on changes in public transit usage among their users in many of the world's major metropolitan areas (*Citymapper, 2020b*, *2021*).

Public transit use is a strong candidate to capture the full range of population-level mobility changes that occurred during the early stages of the COVID-19 pandemic, at least in the major cities where the virus initially hit hardest (*Stier, Berman & Bettencourt, 2021*; *Carozzi, Provenzano & Roth, 2022*). Prior to 2020, public transit use has been shown to be responsive to disruptive events such as disease outbreaks and terrorist attacks, with ridership typically recovering in the months following the event (*Liu, Osorio & Ouyang, 2021*). In early 2020, with the specter of COVID-19 growing, public transit use was affected by a combination of individual behavioral changes (*e.g.*, voluntary reduction in activities), proactive organizational policies (*e.g.*, remote work and virtual classes), government mandates (*e.g.*, lockdowns, stay-at-home orders), as well as service changes (*e.g.*, reduced vehicle capacity, less frequent trips) (*Qi et al., 2023*). In this analysis, we examined the utility of a mobility index based on users of a public transit app as a leading indicator for COVID-19 transmission in 40 global cities during the first wave of the pandemic.

## MATERIALS AND METHODS

### Study design, setting, and study period

We conducted a longitudinal ecological study to estimate the association between public transit mobility reductions and population-level COVID-19 transmission 2 or 3 weeks later. We included 40 cities covered by Citymapper (excluding only Monaco, which did not

record 100 cumulative COVID-19 cases by the end of the study period) (Fig. S1). We considered COVID-19 transmission during the 4 weeks between March 16, 2020 and April 12, 2020.

## Public transit mobility

Citymapper is an app that provides users in select major cities with real-time information on public transportation options, alongside other modes of transport. The Citymapper Mobility Index (CMI) was released on March 19, 2020 (*Citymapper, 2020b*) and included data on 41 cities and metropolitan areas across 23 countries and five continents (*Citymapper, 2020c*). Historical measurements were available back to March 2 for most cities. CMI measures the relative frequency of trips planned within the app compared to a "recent typical usage period" of January 6, 2020–February 2, 2020 (exceptions: February 3, 2020–March 1, 2020 for Paris and December 2, 2019–December 22, 2019 for Hong Kong and Singapore) (*Citymapper, 2020a*). A value of 100% indicates a typical volume of trips planned with the Citymapper app, whereas values above or below 100% indicate increased or decreased trip volumes, respectively.

To compare the evolution of mobility as measured by CMI to the announcement of formal government restrictions, we plotted the value of CMI before and after the first announcement of a major national or sub-national physical distancing intervention, namely gathering restrictions and/or mandatory closures (Table S1).

## COVID-19 transmission

We measured COVID-19 transmission using two different quantities: the weekly case ratio and the effective reproduction number ($R_t$).

To calculate these quantities, we obtained time series of COVID-19 case counts for cities represented in the Citymapper dataset (Table S2). For most cities, it was possible to obtain case data at the local level (*e.g.*, cities, counties, metropolitan areas) corresponding to the area served by the Citymapper app. For a small number of cities where local data were not available, we used national or sub-national data. These data should nonetheless serve as good approximations for the global cities represented in our dataset, as initial transmission of the disease was concentrated in these dense, internationally connected hubs.

We defined the weekly case ratio for a given week as the ratio between cases reported in that week and cases reported in the week prior. A value of 1 indicates the number of cases reported in the present week is unchanged from the prior week.

The effective reproduction number represents the average number of secondary infections a person infected at time $t$ would be expected to generate given that conditions remain unchanged (*Fraser, 2007*), with a value of 1 marking the threshold between the exponential growth ($R_t > 1$) and exponential decay (when $R_t < 1$). We estimated $R_t$ *via* the parametric serial interval method (*Cori et al., 2013*) implemented in the *EpiEstim* (version 2.2-4) (*Cori, 2019*) package in R (version 4.1.3) (*R Core Team, 2021*). We used estimates of the serial interval of the original wild-type virus by *Du et al. (2020)* (mean = 3.96 days, SD = 4.75 days).

For each location, we calculated the weekly case ratio and $R_t$ for the 4 weeks comprising the study period (March 16, 2020 and April 12, 2020). The weekly case ratio could not be calculated during the week beginning March 16 for the cities of Amsterdam, Lyon, and Paris due to insufficient data. Istanbul and Montreal were excluded from the week of March 16 for both the weekly case ratio and $R_t$ because the calculated weekly case ratios exceeded 30. There was a change in how tests were confirmed during this period in the province of Quebec, where Montreal is situated (Rocha, 2020).

## Statistical analysis

We used multilevel linear regression models to estimate the association of public transit mobility reductions with COVID-19 transmission during the initial wave of the pandemic. We fit two sets of models using different outcomes: the logarithm of the weekly case ratio and the effective reproduction number. The primary exposure variable was the weekly mean CMI 2 weeks prior. We chose a 2-week lag to reflect the delay between infection and symptom onset (the incubation period) and the delay between symptom onset and when the results of confirmatory testing are reported (the reporting lag). A meta-analysis using data from the first few months of the pandemic estimated the incubation period of COVID-19 at 6.5 days (Alene et al., 2021). We estimate reporting lag to be between 5–15 days early on in the pandemic, since testing was scarce and rapid diagnostics were unavailable. Some estimates of reporting delays include 6.4 days in Singapore (up to March 17) (Tariq et al., 2020), 4–10 days in Bavaria, Germany (up to April 8) (Günther et al., 2021), and 7–9 days in Japan (up to May 13) (Miyama et al., 2022).

We fit models using four pairs of weeks: comparing average weekly CMI for the weeks beginning March 2, March 9, March 16, and March 23 to the case ratio and reproductive number 2 weeks later (the weeks beginning March 16, March 23, March 30, and April 6, respectively). Most countries implemented their first major physical distancing policies throughout the second week of March (March 9–March 15), so this 4-week period covers time both before and after official government interventions. For most countries in the dataset, this period was a time of growth and/or an early peak in the first wave of the pandemic (Dong, Du & Gardner, 2020).

The multilevel linear regression models used random intercepts to account for the clustering of outcomes by city, nested within countries. Models were fit using *glmmTMB* (version 1.1.8) (Brooks et al., 2017). To adjust for epidemic timing, we fit adjusted models for both outcomes that included a new covariate: days since the 100[th] case in the country. For the first week of case data (March 16), this covariate was negative for some cities. Unadjusted and adjusted models were compared with a likelihood ratio test. A complete mathematical description of the models used in this analysis is available in the Supplemental Methods.

## Sensitivity analyses

To account for uncertainty in the reporting delay, we fit an alternative set of models using a 3-week lag between CMI and outcomes instead of a 2-week lag. In these models, we could
use only 3 weeks of outcome data (March 23, March 30, April 6), since the CMI dataset begins on March 2 for most cities.

Data on imported *versus* locally acquired cases were not available for most cities, so we were unable to adjust for this factor during the analysis. Since many countries implemented international travel restrictions during the third week of March (*Hale et al., 2021*), cases reported at least 2 weeks later should be much less likely to represent imported cases than cases reported before this time. As a sensitivity analysis, we re-ran models excluding outcome data from the third and fourth weeks of March (beginning March 16 and March 23).

To assess whether the association of CMI with outcomes differed across the 4 weeks included in the 2-week lag models (both unadjusted and adjusted for epidemic timing), we re-fit these models with an additional interaction term between CMI and week and compared them to the original models using a likelihood ratio test.

### Ethics
This study was exempt from review because it used exclusively publicly available data.

## RESULTS
For 40 global cities, we analyzed the association between a public transit mobility index and two measures of COVID-19 transmission over a 4-week period in 2020 assuming a 2-week lag (outcome: March 16–April 12, mobility: March 2–March 29). We had complete data for every week of outcome data except the first, where only 35/40 cities (weekly case ratio analysis) and 38/40 cities (effective reproduction number analysis) had data.

### Public transit mobility
Nearly all cities experienced substantial reductions in public transit mobility during the month of March (March 2: median = 100%, first quartile ($Q_1$)–third quartile ($Q_3$) = 94–108%; March 29: median = 10%, $Q_1$–$Q_3$ = 6–15%) (Fig. 1). Decreases were less pronounced in Hong Kong, Seoul, Tokyo, and Milan, where public transit mobility was already substantially reduced at the beginning of the month. Cities in Europe, Australia, and the Americas showed similar patterns in public transit mobility reductions generally coinciding with the dates of national or sub-national physical distancing measures, including restrictions on public gatherings or business closures (Table S1).

### Weekly case ratio
The weekly value of the mobility index was associated with the logarithm of the weekly case ratio 2 weeks later (Fig. 2, Fig. S2). A 10% lower mean mobility index was associated with a 12.3% lower weekly case ratio 2 weeks later ($\exp(\beta)$ = 0.877; 95% confidence interval (CI): [0.859–0.896]) (Table 1). This model explained a portion of the variance in weekly case ratios (conditional $R^2$ = 0.595). When the model was adjusted for days since the 100th case in the country (a measure of epidemic timing), the association was attenuated to 7.8% ($\exp(\beta)$ = 0.922; 95% CI: [0.885–0.961]) and variance explained increased slightly (conditional $R^2$ = 0.641). Being an additional 10 days since the 100th case in the country

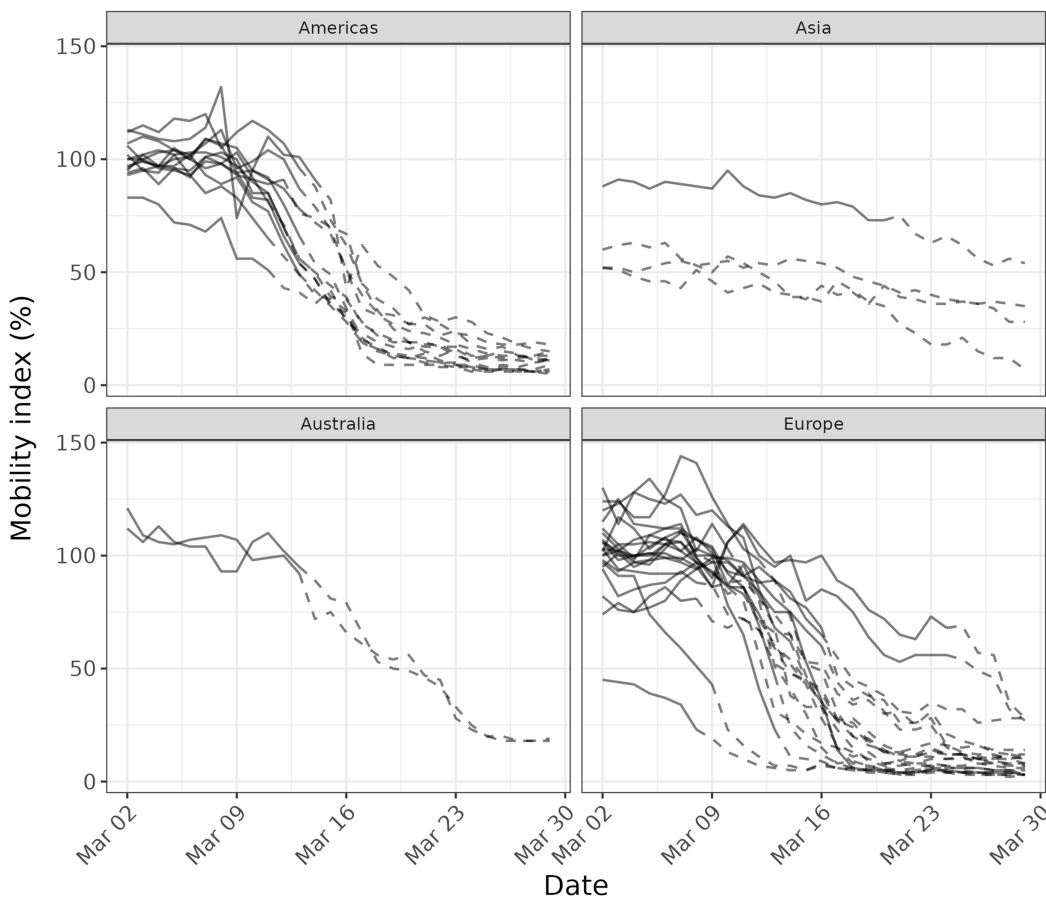

**Figure 1** Mobility index in 40 cities from March 2–March 29, 2020, before (solid lines) and after (dashed lines) the first major national or sub-national physical distancing intervention was announced.

was associated with a 20.8% lower weekly case ratio ($\exp(\beta) = 0.792$; 95% CI: [0.668–0.938]).

Since reporting delay is unknown and variable across geographies, we also ran a model using a 3-week lag for CMI, which produced slightly weaker associations when compared to the full model (Table 1, Fig. S3). The sensitivity analysis excluding the first 2 weeks of outcome data (which have a higher probability of representing imported cases, rather than local transmission) produced comparable results to the full model (Table S3). In the sensitivity analysis of adding an interaction between CMI and week, the likelihood ratio tests did not favor the interaction model in either the unadjusted or adjusted analyses (unadjusted $p = 0.400$, adjusted $p = 0.911$).

## Effective reproduction number

The weekly value of the mobility index was associated with the effective reproduction number 2 weeks later (Fig. 3, Fig. S4). A 10% lower mean mobility index was associated with a decrease in the effective reproduction number 2 weeks later ($\beta = -0.058$; 95% CI:

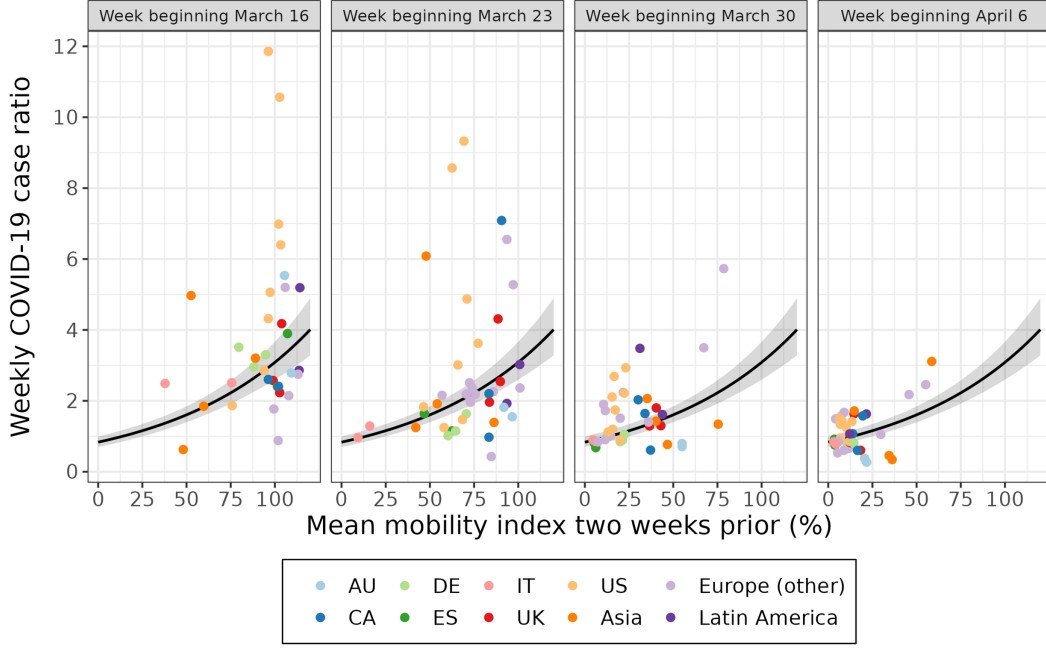

**Figure 2** **The association between the weekly case ratio and the mean mobility index 2 weeks prior for 40 cities over 4 weeks.** Each point represents a week of data for a given city. Each week of data (labelled according to the week the outcome was measured) is plotted separately for a multilevel model fit on all 4 weeks of data. The 95% confidence interval of the fitted value is shown in grey. The first week of data (March 16) excludes Amsterdam, Istanbul, Lyon, Montréal, and Paris. AU = Australia; CA = Canada; DE = Germany; ES = Spain; IT = Italy; UK = United Kingdom; US = United States.

**Table 1** **Estimated model coefficients (with 95% confidence intervals) for the association between a 10% decrease in the mobility index and the weekly growth rate and effective reproduction number assuming a lag of 2 or 3 weeks.**

| | Weekly case ratio ($\exp(\beta)$ = ratio of current week to previous week) | | | |
| --- | --- | --- | --- | --- |
| | **2-week lag** | | **3-week lag** | |
| | **Unadjusted** | **Adjusted** | **Unadjusted** | **Adjusted** |
| Mobility[a] | 0.877 (0.859, 0.896) | 0.922 (0.885, 0.961) | 0.899 (0.877, 0.921) | 0.938 (0.896, 0.982) |
| Days since 100th case[b] | | 0.792 (0.668, 0.938) | | 0.791 (0.635, 0.985) |
| Marginal $R^2$ | 0.439 | 0.477 | 0.312 | 0.361 |
| Conditional $R^2$ | 0.595 | 0.641 | 0.587 | 0.642 |
| Likelihood ratio test | | $\chi^2(1) = 7.380, p = 0.007$ | | $\chi^2(1) = 4.472, p = 0.034$ |
| | **Effective reproduction number ($\beta = \Delta R_t$)** | | | |
| | **2-week lag** | | **3-week lag** | |
| | **Unadjusted** | **Adjusted** | **Unadjusted** | **Adjusted** |
| Mobility[a] | −0.058 (−0.068, −0.048) | −0.032 (−0.052, −0.013) | −0.046 (−0.056, −0.035) | −0.022 (−0.043, −0.002) |
| Days since 100th case[b] | | −0.121 (−0.202, −0.040) | | −0.132 (−0.233, −0.031) |
| Marginal $R^2$ | 0.413 | 0.455 | 0.286 | 0.353 |
| Conditional $R^2$ | 0.587 | 0.647 | 0.623 | 0.701 |
| Likelihood ratio test | | $\chi^2(1) = 8.749, p = 0.003$ | | $\chi^2(1) = 6.625, p = 0.010$ |

**Note:**
Model coefficients are presented with and without adjustment for days since 100th case. Models include outcome data from March 16–April 12 (2-week lag) or March 23–April 12, 2020 (3-week lag) for 40 cities.
[a] Coefficient for a 10% decrease in the mobility index.
[b] Coefficient for a 10-day increase since the 100th reported case in the country.
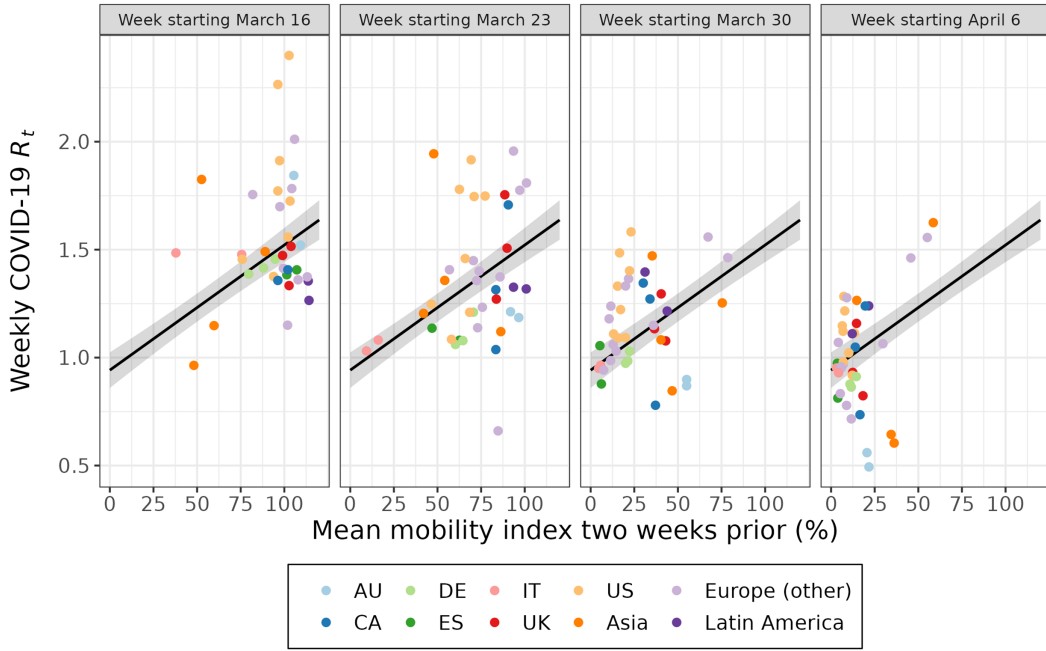

**Figure 3** **The association between the effective reproduction number and the mean mobility index 2 weeks prior for 40 cities over 4 weeks.** Each point represents a week of data for a given city. Each week of data (labelled according to the week the outcome was estimated) is plotted separately for a multilevel model fit on all 4 weeks of data. The 95% confidence interval of the fitted value is shown in grey. The first week of data (March 16) excludes Istanbul and Montréal. AU = Australia; CA = Canada; DE = Germany; ES = Spain; IT = Italy; UK = United Kingdom; US = United States.

[−0.068 to −0.048]) (Table 1). This model explained a portion of the variance in the effective reproduction number (conditional $R^2$ = 0.587). When the model was adjusted for days since the 100th case in the country, the association was attenuated ($\beta$ = −0.032; 95% CI: [−0.052 to −0.013]) and variance explained increased slightly (conditional $R^2$ = 0.647). Being an additional 10 days since the 100th case in the country was associated with a decrease in the effective reproduction number ($\beta$ = −0.121; 95% CI: [−0.202 to −0.040]).

Using a 3-week lag for CMI produced a slightly weaker strength of association (Table 1, Fig. S5). The sensitivity analysis excluding the first 2 weeks of outcome data (which have a higher probability of representing imported cases, rather than local transmission) produced roughly comparable results to the primary analysis, based on 2-week lag (Table S3). In the sensitivity analysis of adding an interaction between CMI and week, the likelihood ratio tests did not favor the interaction model in either the unadjusted or adjusted analyses (unadjusted $p$ = 0.539, adjusted $p$ = 0.958).

## DISCUSSION

Public transit mobility, as measured by data from users of a public transit app in 40 cities, dropped by 90% over the course of March 2020. In most cities, the Citymapper Mobility Index began dropping prior to official government actions, pointing to voluntary reductions in mobility from individuals and non-governmental organizations. We found these reductions in the mobility index were strongly associated with reduced COVID-19 transmission 2 to 3 weeks later. The CMI-only (unadjusted) models for both weekly case

ratio and effective reproduction number explained nearly 60% of variance in COVID-19 transmission. The addition of a variable for epidemic timing (adjusted model) somewhat attenuated the observed associations while only slightly increasing the variance explained. This is not surprising, as epidemic timing is closely related to the implementation of measures to limit mobility; for example, Italy, which experienced an earlier outbreak than other European countries, had a substantially lower CMI at the beginning of March. Together, these results suggest that public transit mobility was a good leading indicator for COVID-19 transmission in 40 global cities during the first wave of the pandemic, a period marked by a scarcity of such metrics.

These results add to the substantial body of literature supporting the use of human mobility data for a variety of applications related to COVID-19, such as forecasting disease burden and evaluating the effectiveness of non-pharmaceutical interventions (*Kraemer et al., 2020*; *Askitas, Tatsiramos & Verheyden, 2021*; *Nouvellet et al., 2021*; *Leung, Wu & Leung, 2021*; *Brown et al., 2021*; *Ilin et al., 2021*; *Soucy et al., 2021*; *Alessandretti, 2022*). It also contributes to the literature evaluating novel surveillance metrics that showed promise early in the pandemic, such as Internet searches for loss of taste and smell (*Brunori & Resce, 2020*; *Asseo et al., 2020*). Like this other metric, public transit mobility diminished in utility over time, for a variety of reasons. The elasticity of public transit mobility was bounded by factors such as remote work, crime, and reliability, leaving ridership depressed years after the initial outbreak (*Zukowski, 2023*). As the CMI is based on the volume of trips planned in the app, less reliable public transit service could also make users of the app more likely to explicitly plan their trip than before, exaggerating the recovery of public transit use. Additionally, research has found that the association between mobility and COVID-19 transmission has changed over time (*Nouvellet et al., 2021*; *Alessandretti, 2022*).

There are several limitations with the measurements of COVID-19 transmission and mobility used in this analysis. For example, a mobility index cannot directly capture non-contact limiting measures, such as masking, contact tracing, and vaccination. However, these measures became more prominent later in the pandemic (*e.g.*, U.S. CDC recommended against community masking until April 2020 and WHO until June 2020 (*Zhang et al., 2021*); YouGov surveys in the U.S. reported 11% mask wearing on March 23 but 63% by April 27 (*YouGov, 2022*)). These non-contact limiting measures are likely important to explaining the trajectories of the four Asian cities in the CMI dataset, which are notable outliers in having among the highest public transit mobility by the end of the study period. Much has been written of the ability of countries like South Korea, Japan, and Taiwan to control their early COVID-19 outbreaks without the same kinds of broad, mandatory restrictions on mobility employed elsewhere (*Park, Choi & Ko, 2020*; *Chen et al., 2021*; *Jae Moon et al., 2021*).

A limitation with respect to case data is that case ascertainment likely changed over time, introducing error into our outcome variables (*Noh & Danuser, 2021*). Additionally, our estimates of the effective reproduction number did not account for imported cases, but a sensitivity analysis restricting the analysis to the time period following when most

countries shut down international travel showed broadly similar results to the main analysis.

## CONCLUSIONS

Our analysis showed that a novel public transit mobility metric served as a useful leading indicator of COVID-19 transmission during the first wave of the COVID-19 pandemic. While the utility of a metric based on public transit mobility is likely to diminish over time, this measure effectively captured epidemic trends during the early stages of the pandemic, when few other metrics for COVID-19 transmission were available. This study serves as an example of how "big data" from industry can be used to augment responses to important public health challenges, provided these applications are developed within a transparent and ethical framework. We hope our research helps to make the case for continued industry support of "data for good" to inform and refine public health strategies.

### Funding
The authors received no funding for this work.

### Competing Interests
The authors declare that they have no competing interests.

### Author Contributions
- Jean-Paul R. Soucy conceived and designed the experiments, performed the experiments, analyzed the data, prepared figures and/or tables, authored or reviewed drafts of the article, and approved the final draft.
- Shelby L. Sturrock conceived and designed the experiments, authored or reviewed drafts of the article, and approved the final draft.
- Isha Berry conceived and designed the experiments, authored or reviewed drafts of the article, and approved the final draft.
- Duncan J. Westwood conceived and designed the experiments, authored or reviewed drafts of the article, and approved the final draft.
- Nick Daneman conceived and designed the experiments, authored or reviewed drafts of the article, and approved the final draft.
- David Fisman conceived and designed the experiments, authored or reviewed drafts of the article, and approved the final draft.
- Derek R. MacFadden conceived and designed the experiments, authored or reviewed drafts of the article, and approved the final draft.
- Kevin A. Brown conceived and designed the experiments, analyzed the data, authored or reviewed drafts of the article, and approved the final draft.

### Data Availability
All data and code necessary to reproduce the results are available at Zenodo: Soucy, J.-P. R., Sturrock, S. L., Berry, I., Westwood, D. J., Daneman, N., Fisman, D., MacFadden, D. R., & Brown, K. A. (2024). jeanpaulrsoucy/covid-19-mobility. Zenodo. https://doi.org/10.5281/zenodo.11062751.

## Supplemental Information

Supplemental information for this article can be found online at http://dx.doi.org/10.7717/peerj.17455#supplemental-information.

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
