# Peer review of "Public transit mobility as a leading indicator of COVID-19 transmission in 40 cities during the first wave of the pandemic"

_PeerJ, doi:10.7717/peerj.17455_

## Round 0.1 · original submission · Major Revisions

Though reviewers provided very positive feedback on the quality of your manuscript and results, they nevertheless raised some major concerns that need to be addressed. Please provide a thorough response to the reviewers' comments as the manuscript will be returned to them for further evaluation.

Reviewer 1 ·

Basic reporting

Clear and unambiguous English: I think this was fairly well written, but there were times when they referred simply to general "mobility" when they really meant "public transit mobility data" (i.e., from the CMI), and since they refer to both in this article, I feel like it is important distinction to make. If they were concerned about length or wordiness, they could have referred to the "CMI" as a "short cut" term instead of "mobility", since that was the only public transit index that was used.

Literature references: There were a few places where I think the authors should have cited publications to back up statements, including lines 68-72 where they assert that public transit use was affected by a series of factors, and while some of that may be inferred, there are references to back this up (like https://www.ncbi.nlm.nih.gov/pmc/articles/PMC8611694/). Also, they do not cite any of their sources for the COVID surveillance datasets (lines 106-109).

Professional article structure: I think the tables and figures were informative (though the supplementary figures would have been improved by captions), but no raw data (or even derived data) was shared. I tried to find past CMI datasets (since that index is no longer available) but it did not seem to be available from the link they provided. Since they did not cite their sources for the COVID-19 case counts, that was not findable, either.

Experimental design

Research question well defined: The authors discuss how there were not very many indicators for COVID-19 transmission early in the pandemic, and I agree public transit mobility sounds like it was a good early leading indicator in a correlative way (and the authors do demonstrate this well, at a high level) but that seems kind of intuitive. The authors do at least confirm what one might have suspected.

Rigorous investigation: The authors say that, over time, public transit became less useful as an indicator because of the decrease in ridership -- it might have been interesting if they had examined the correlation past the four week study period to see when public transit mobility began to wane as a good indicator. They also alluded to differences in pandemic status (and reporting) influencing results (eg, Asia: 185-187; Italy: 238-240; reporting delays: 143-145) -- perhaps if they had compared the results based on pandemic progression instead of just adhering to the 4 weeks they selected (eg, matching cities based on the week of first cases per city rather than starting from a specific calendar date) they may have found that the index was an even better indicator.

Methods: Methods were described at a relatively high level, and not with sufficient detail and information to replicate. They mentioned that they relied on local COVID-19 counts for their COVID-19 measures (lines 106-109), but there were a couple of global COVID-19 surveillance sites (like the COVID-19 Dashboard by the Center for Systems Science and Engineering (CSSE) at Johns Hopkins University (JHU)) that might have provided the data in a more uniform/comparable/consistent way, which may also have improved their findings.

Validity of the findings

All underlying data: No underlying data was provided, the COVID-19 data sources were not cited, and the CMI data is not readily available from the cited location.

Conclusions are well stated: I would say that their conclusions mostly support their hypothesis. There was one assertion they made multiple times (169-172; 204-205; 221-222) to the effect of "Cases reported during the excluded weeks are much more likely to represent imported cases than cases reported during later weeks" (they cited Hale, et al. 2021 the first time they made this assertion, but Hale, et al. don't really make this assertion), and while it might be inferred that that is the case (eg, that there would be fewer imported cases before travel restrictions are imposed, which may also coincide with the second week of the study period), that does not mean that all cities would be affected to similar extent or that imported cases were a significant factor (e.g., some cities may see less international travel in general (or from affected countries), some may have imposed travel restrictions earlier than the four week period, etc.)

Additional comments

Overall, I found this a very readable article. I know PeerJ is not assessing the article based on novelty, but I do think the idea of using a public transit index as an indicator of COVID-19 is rather interesting, especially since there is no evidence of causation here, just correlation. The authors also point out -- rightly -- that there weren't many good indicators at the beginning of the pandemic.

Since the studied period is four years ago, though, it would be interesting if they had gone into more detail on how this information could be used in the future, or to gather more insights on how it performed as an index over time. I just feel like, since it is too late to use the CMI as a COVID-19 indicator now, it would be nice to know how we might use this information to help us plan (or assess) in the future.

·

Basic reporting

The writing is generally good.

The assumptions seem valid based on the cited literature, which is basically consistent with my understanding from reading other sources.

The figures are acceptable, but Figure 1 should be split into multiple subfigures, one for each region.

The objective of the study was to demonstrate the utility (or lack thereof) of using aggregate public transport data as a leading indicator of COVID transmission, and the results are sufficient to address the objective. The quality of the 'leading indicator' could be analysed and discussed more thoroughly, for example, out-of-sample prediction could be used to provide a more robust measure of the usefulness of the indicator (beyond post-hoc model fitting).

Experimental design

The questions were certainly relevant during the early stages of the COVID-19 pandemic, and I think the article has value in that retrospective context.

To me the technical standard appears adequate for the study's scope and purpose. However, I am not qualified to review the technical details of of the multi-level linear model used in this work. A more thorough mathematical description of the modelling would be helpful for non-expert readers to understand the approach. I also think a justification for the statistical method used should be provided and discussed.

Validity of the findings

The findings appear valid, though see above about demonstrating the usefulness of the model for prediction.

The underlying data is available in the associated repository (I suggest making a stable release on Zenodo).

The conclusions are in-scope.

Reviewer 3 ·

Basic reporting

This work attempts to contribute to the existing literature on the association of mobility and COVID-19 transmission from the perspective of public transit usage. Using CityMapper's mobility index across 40 cities during the critical period of COVID-19, the authors show the association between public transit mobility and COVID-19 weekly case rate and reproduction number. Their analysis shows that public transit mobility is generally a good indicator of these disease dynamics.

The paper is easy to read, with a clear research motive and coherent experiment designs. The results are shown with generally appropriate figures. However, in my opinion, the paper lacks its referenced literature, which in my opinion is the most important thing to address before the paper is ready to be accepted.

As mobility-COVID transmission is a rich literature, there exist other forms of mobility indicators that were available during the study period examined by the authors (e.g. CBG-POI occupancy[1], interaction metrics[2]). A comprehensive analysis of all these mobility indicators along with public transit mobility and a comparison between them would be very interesting and a useful contribution to the literature. However, I do understand that some of these datasets might be private and not so easily available and hence comparison may be infeasible. Therefore, at the very least, I would like to see a more thorough discussion surrounding various kinds of mobility indicators in literature and the justification of using public transit mobility, if comparison is not possible.

1. Chang, Serina, et al. "Mobility network models of COVID-19 explain inequities and inform reopening." Nature 589.7840 (2021): 82-87.

2. Mehrab, Zakaria, et al. "Evaluating the utility of high-resolution proximity metrics in predicting the spread of COVID-19." ACM Transactions on Spatial Algorithms and Systems 8.4 (2022): 1-51.

Experimental design

No Comment

Validity of the findings

Comments on Fig 2 & 3:

1) The leftmost panel seems to have fewer than 40 cities. Is it because some plots are completely overlapped by others? If that is not the case, please mention why there are fewer points.

2) I think there is an alternative way to draw these 2 figures. These two figures do not show that the association between public transit mobility and COVID dynamics is not so obvious for some cities. For example, in Figure 2, on March 30, no city in Asia has a case ratio above 2 and the mean mobility index is around 75% at best. However, the next week there is a city with a Case ratio of around 3 although the max mobility index is below 50% for the cities of Asia. Maybe the point I am looking at is for a city in Asia that had a lower mobility index and lower case ratio in the prior week, but it should be presented more clearly. Maybe a large figure in the supplementary file showing the trend between mobility and case ratio for all 40 cities will better help show the cities with anomalies. If there are indeed such anomalies, a discussion surrounding such cities should also be presented.

---

## Round 0.2 · Minor Revisions

Please, address the minor revisions suggested by Reviewer #1

Reviewer 1 ·

Basic reporting

Clear and unambiguous professional English was used, though there were a few minor grammar hiccups:
* Ln 75: “as well service changes” should be “as well as service changes”
* Ln 109: remove comma between “areas)” and “corresponding”
* Ln 271: "such masking" should be "such as masking"

Literature references were improved.

Professional article structure, figures, etc:
* I still could not see a caption for S1 when looking at it, but maybe that is a limitation of the PeerJ site.
* Links to the raw data were provided in one of the supplementary tables, though I could not get the Yandex links to load.

Self-contained with relevant results to hypotheses:
I think so. I had one doubt on this related to the conclusion, but I describe that below.

Experimental design

No comment. I found the Supplementary methods section useful.

Validity of the findings

Conclusion:
In “Discussion” section of the abstract, would recommend getting rid of the last sentence: “Continued industry support is vital for developing innovative metrics to confront pressing public health issues.” It seems generic, not specific to this article, and seems especially out of place given that the previous sentence says that public transit mobility ceased to be useful as an indicator since ridership is still depressed since the pandemic.

Similarly, the sentence in the Conclusion, “Innovative applications of this sort of ‘big data’ held by many firms, when done within a transparent and ethical framework, hold great promise for enhancing societal responses to many health challenges.” I feel like this study is an example of how industry big data could be used as a metric to evaluate a health challenge, but this article does not prove that, broadly speaking, industry big data is useful in general. I think I would soften this a bit, perhaps phrasing it more in terms of, “This study serves as an example of the innovative potential for industry big data in enhancing societal responses to many health challenges, provided such applications are developed within a transparent and ethical framework.”

Additional comments

This is an improvement over the previous draft.

·

Basic reporting

The Authors adjusted Figure 1 according to my suggestion. The revised version is an improvement.

The authors did not make changes to address my main suggestion regarding further analysis of public transport use as a predictor per-se, but did provide a rebuttal that I'm prepared to accept, although I don't entirely understand the final statement: "An alternative strategy could have been to hold out “test cities” from the initial dataset, but this would be at cross purposes with our objective of investigating the general phenomenon of mobility reductions across all cities represented in the CMI dataset" - I don't see why testing the model with out-of-sample data is at cross-purposes, you can still report the model that uses the full dataset, but it would be helpful to see a demonstration of how the model performs in test cases (i.e., leaving out a single city from the analysis and then testing the quality of the predictive model fit to the rest of them). There are no drawbacks to doing this that I can see (other than the additional work involved), so I don't find the statement above to be convincing.

That said, the findings are still adequate to make a useful contribution to the literature so I'm inclined to accept this as a limitation of the study.

Experimental design

The authors addressed my suggestions. I appreciate the additional method description in the supplement, that made the it much more straight-forward to understand.

Validity of the findings

no further comments

Additional comments

no additional comments.

Reviewer 3 ·

Basic reporting

No Comment

Experimental design

No Comment

Validity of the findings

No Comment

Additional comments

The authors have addressed my comments sufficiently. I have no further concerns.

---

## Round 0.3 · accepted · Accept

Thank you for thoroughly addressing all the reviewer's comments.